# Hsp70–Bag3 Module Regulates Macrophage Motility and Tumor Infiltration via Transcription Factor LITAF and CSF1

**DOI:** 10.3390/cancers14174168

**Published:** 2022-08-28

**Authors:** Lena Avinery, Valid Gahramanov, Arkadi Hesin, Michael Y. Sherman

**Affiliations:** Department of Molecular Biology, Ariel University, Ariel 407000, Israel

**Keywords:** tumor-associated macrophages, macrophage invasion, LITAF, chaperone-mediated autophagy

## Abstract

**Simple Summary:**

Patients’ normal cells, such as lymphocytes, fibroblasts, or macrophages, can either suppress or facilitate tumor growth. Macrophages can infiltrate tumors and secrete molecules that enhance the proliferation of cancer cells and their invasion into neighboring tissues and blood. Here, we investigated the mechanism of action of a novel small molecule that suppresses the infiltration of macrophages into tumors and demonstrates potent anticancer activity. We identified the entire pathway that links the intracellular protein Hsp70, which is inhibited by this small molecule, with the macrophage motility system. This study will lay the basis for a novel approach to cancer treatment via targeting tumor-associated macrophages.

**Abstract:**

The molecular chaperone Hsp70 has been implicated in multiple stages of cancer development. In these processes, a co-chaperone Bag3 links Hsp70 with signaling pathways that control cancer development. Recently, we showed that besides affecting cancer cells, Hsp70 can also regulate the motility of macrophages and their tumor infiltration. However, the mechanisms of these effects have not been explored. Here, we demonstrated that the Hsp70-bound co-chaperone Bag3 associates with a transcription factor LITAF that can regulate the expression of inflammatory cytokines and chemokines in macrophages. Via this interaction, the Hsp70–Bag3 complex regulates expression levels of LITAF by controlling its proteasome-dependent and chaperone-mediated autophagy-dependent degradation. In turn, LITAF regulates the expression of the major chemokine CSF1, and adding this chemokine to the culture medium reversed the effects of Bag3 or LITAF silencing on the macrophage motility. Together, these findings uncover the Hsp70–Bag3–LITAF–CSF1 pathway that controls macrophage motility and tumor infiltration.

## 1. Introduction

Hsp70 (HSPA1A) is a stress-inducible molecular chaperone that plays important roles in protein folding, degradation, and cell survival [1]. Multiple observations suggest that Hsp70 is involved in cancer development, including the presence of elevated levels of Hsp70 in tumor tissues and the correlation of Hsp70 levels with cancer stage and drug resistance [2]. Furthermore, knockout of Hsp70 prevents the development of tumors in the model of Her2-positive breast cancer in mice [3]. These effects are associated with enhanced oncogene-induced senescence in breast epithelial cells in Hsp70 knockout animals. Works with other mouse cancer models, e.g., breast cancer caused by the polyomavirus middle T oncogene [4], colon cancer [5], or carcinogen-induced hepatocellular carcinoma [6], indicated that Hsp70 controls multiple stages of cancer development, including initiation, progression, and metastasis. These effects are associated with changes in multiple signaling pathways. Multiple publications demonstrated that depletion of Hsp70 alters activities of major cancer-related signaling pathways, including Akt, Hif, p53/p21, MAP kinases, NF-kB, Myc, and probably others [2,3,7,8]. In its function in cancer signaling, Hsp70 collaborates with a co-chaperone Bag3, which is co-elevated with Hsp70 in many tumor types [9,10]. Indeed, Bag3 was shown to regulate pathways controlled by Hsp70 [11,12,13], including Akt [14], NF-kB [15], BRAF [16,17], Mcl-1 [18,19], Bcl-2 and Bcl-XL [20] and others [8]. Bag3 also plays a role in the regulation of cell adhesion and cell motility [21,22,23,24,25]. 

A series of small molecules (e.g., JG-98) has been developed that bind to the ATPase domain of Hsp70 and disrupt its interaction with Bag3 [26]. These molecules have been used as tools to probe the effect of the Hsp70–Bag3 complex on cell signaling and other processes [26]. Indeed, JG-98 displays potent anticancer effects both in vitro and in vivo by modulating many of these signaling pathways [27]. 

Recently, it was shown that Hsp70 plays a critical role in the tumor microenvironment by controlling the infiltration of macrophages into tumors [28]. Indeed, either depletion of Hsp70 or its inhibition by JG-98 in macrophages led to suppression of their motility and infiltration into the tumor site [29]. Since JG-98 blocks the interaction between Hsp70 and Bag3, we suggested that Bag3 may also be involved in the control of macrophage motility. This possibility was in line with previous reports that Bag3 depletion suppresses cell migration [24].

Tumor-associated macrophages (TAMs) contribute to tumor growth and progression by promoting stimulating angiogenesis, supporting cancer stem cells and metastasis, and taming adaptive immunity [30]. TAMs also contribute to tumor relapse by initiating regenerative programs, which result from macrophages’ function in wound healing [31,32]. Chemoattractants secreted by macrophages that control their recruitment to the tumor site include chemokines, e.g., CCL2 or CCL5, and cytokines, e.g., CSF-1 and members of the VEGF family [33,34]. Among them, CSF1 plays a special role in differentiation towards tumor-promoting “M2-like” phenotype [35,36], and regulation of macrophage motility [37,38,39,40]. CSF-1-secreting tumor cells and EGF-secreting TAMs stimulate each other, leading to the migration of both cell types together towards blood vessels [41,42,43,44]. Indeed, co-migration of TAMs and tumor cells along collagen fibers within tumors have been reported [45]. In line with these findings, high CSF-1 levels coincide with TAM clustering at invading fronts of breast cancers [46,47,48,49].

The transcription factor lipopolysaccharide TNFα factor (LITAF) is highly expressed in macrophages and enhances the expression of TNF-α, CCL2, and other inflammation-promoting factors [50]. In this function, LITAF co-activates these promotors together with NF-kB [51]. The N-terminus of LITAF contains two PPXY (PY) motifs that can bind to WW domains. Via this interaction, LITAF can bind oxidoreductase (WWOX), a family of HECT domain ubiquitin ligases, e.g., NEDD4, Itch WWP1/2, Smurf1/2 [52], and TSG101 [53,54,55,56]. Importantly, Bag3 contains the WW-domain and was identified as one of the major LITAF binding partners using BioGrid data, high-throughput complex pull downs, and BioID assay [57].

In an unpublished Ph.D. thesis, LITAF was shown to bind to the Bag3 WW domain via its first PPXY region [57]. Here, we hypothesized that the recruitment of TAMs into the tumor site is regulated by the Hsp70–Bag3 complex via LITAF.

## 2. Materials and Methods

### 2.1. Cell Culture 

Human monocyte THP-1 cells and H1975 were maintained in culture in RPMI1640 (Gibco #21875-034) culture medium containing 10% heat-inactivated FBS (Gibco #26140-079, MA, USA) and supplemented with 10 mM Hepes (Biological industries #03-025-1B, Beit HaEmek, Israel) and 1 mM pyruvate (Biological industries #03-042-1B, Beit HaEmek, Israel). The 293T cells were maintained in Dulbecco’s modified Eagle’s medium (Sartorius #01-055-1A, Beit HaEmek, Israel) supplemented with 10% FBS. All cultures were supplemented with L-glutamine (Sartorius #03-020-1B), as well 0.1 mg/mL penicillin (Sartorius #03-031-1B) and kept at 37 °C and 5% CO_2_. THP-1 monocytes were differentiated into macrophages by incubation with 100 ng/mL phorbol 12-myristate 13-acetate (PMA, Sigma aldrich # P8139, Burlington, MA, USA) for 24 h. 

### 2.2. siRNA Transfection and Treatments

THP1 cells were transfected with siRNA using HiPerFect Transfection Reagent (QIAGENh, Hilden, Germany) according to the manufacturer’s instructions. Thus, for a well on a 35 mm dish, we plated 1 × 10^6^ cells/mL, then we mixed 24.5 μL of the reagent with 3 μL of 5 μM siRNA in 378 μL of OptiMEM and added the mixture to 400 μL of a cell suspension in the well. After 72 h, the transfection was stopped, and the cells were plated for an experiment conducted on the same day. We used the following siGENOME siRNAs purchased from Dharmacon (Lafayette, CO, PA, USA): standard personalized sequence Bag3 5′-UCAGGAAGGUUCAGACCAUUU-3′ or 3′ UTRBag3 5′-GCCAUAGGAAUAUCUGUAUUU-3′ or Smartpool sequences for LITAF for normal silencing. For plasmid transfection, we used lipofectamine 3000 (Thermo Fisher Scientific, #L3000001 Waltham, MA, USA) and followed the manufacturer’s protocol. 

### 2.3. Cell lysis and Analysis 

A total of 1 × 10^6^ cell/mL THP1 cells in 35 mm or 60 mm dishes were lysed with lysis buffer (40 mM HEPES, pH 7.5, 50 mM KCl, 1% Triton X-100, 2 mM dithiothreitol, 1 mM Na3VO4, 50 mM β-glycerophosphate, 50 mM NaF, 5 mM EDTA, 5 mM EGTA), and supplemented with proteasome inhibitor cocktail (Sigma aldrich) and PMSF before the use. Samples were adjusted to have equal concentrations of total protein and subjected to PAGE followed by immunoblotting, as described previously [58]. 

For pull-down analysis of Bag3-associated proteins, for each condition, transfected 293T cells from 2100 mm dishes were washed with DPBS, fixed with 1.2% formaldehyde for 10 min at room temperature, then Tris-HCl (pH 7.4) was added to 50 mM final concentration which was followed by a wash with 50 mM Tris-HCl (pH 7.4) in DPBS. All the following steps were performed at 4 °C. The cells were lysed in DPBS (Corning) supplemented with 30 mM NaCl, 10 mM Hepes (pH 7.4), 1.5 mM MgCl2, 0.5% Triton X-100, 5% glycerol, 10 mM imidazole, 1 mM PMSF, and protease and phosphatase inhibitor cocktails (Sigma P8849 and 4906845001, respectively). The lysates were passed three times through a syringe (21G needle) and clarified by centrifugation for 7 min at 16,000 g. The supernatants were adjusted to have equal concentrations of total protein and loaded on 15 μL of HisPur^TM^ Ni-NTA Resin (Thermo Fisher Scientific #88221). After incubation for 40 min, the flow through was allowed to pass through the beads twice more, and the beads were washed five times with DPBS (#SH30028.02 Corning, New York, NY, USA) supplemented with 146 mM NaCl, 20 mM Tris-HCl (pH 8.0), 0.5% Triton X-100, 5% glycerol, and 15 mM imidazole. The His-tagged Bag3 along with associated proteins was eluted with 300 mM imidazole in 50 mM Na (PO4) (pH 6.8) and 300 mM NaCl.

### 2.4. Lysosomal Inhibition Using Chloroquine and NH4Cl 

PMA differentiated THP1 cells were incubated with 100 µM chloroquine for 4 h or 50 mM NH4Cl for 30 minutes. Cells were lysed and proteins were subjected to PAGE followed by immunoblotting as described in the above section.

### 2.5. Microscopy 

Microscopy was performed using control and Bag3 or LITAF siRNA silenced PMA differentiated THP1 cells plated in a concentration of 8 × 10^3^ cells/well in a total volume of 200 µL. The assay was carried out as previously described by Baldan et al. [59]. The cells were fixed with a solution of 5% formaldehyde in PBS at room temperature for 15 min, washed once with PBS, and permeabilized with 0.2% Triton X-100 in PBS for 10 min at room temperature. The cells were blocked for 1 h in blocking solution (5% BSA (*w*/*v*) in PBST) at room temperature, rinsed three times, and washed three times for 5 min with PBST. Next, a solution of primary antibodies (1:200) in the blocking solution was added to the cells, which were incubated overnight at 4 °C. On the next day, the cells were rinsed three times and washed six times for 5 min with PBST. A solution of secondary antibodies (1:500) in blocking solution was added to cells and left to incubate for 1 h at room temperature, after which the cells were washed as previously and DAPI was added at the concentration of 1:5000 followed by 3 washes with PBST. Fluorescence microscopy was performed with WiScan^®^ Hermes High Content Imaging System (IDEA Bio-Medical, Rehovot, Israel). The cells were plated in a black 96-well plate with a transparent bottom. Fifty-two images per well were automatically acquired, which corresponds to thousands of cells per sample. The images were taken at room temperature using a 20× objective, utilizing mainly the red (Ex. 560/32, Em. 607/36), blue (Ex. 390/22, Em. 440/40), and green (Ex. 485/25, Em. 525/30) channels. Image quantification was performed with the WiSoft^®^ Athena software Translocation Application (IDEA Bio-medical, Rehovot, Israel). The software application performs automated, multiplexed image analysis by processing the blue fluorescence channel for nuclei detection using watershed segmentation, which was calibrated with parameters that allow separation between adjacent objects; the red channel for detection of cytoplasm using the nucleus as a seed in seeded watershed analysis and the green channel to quantify for the amount of protein in each compartment. Throughout different experiments, these parameters were kept equal to maintain homogeneity during the analysis. The program allows adjusting the following parameters: Nucleus Smooth, Nucleus Background Subtraction, Nucleus Intensity Threshold, Nucleus Max Patch Size, Nucleus Maximum Area, Nucleus Minimum Area, Cell Smooth, Cell Background Subtraction, and Cell Intensity Threshold [60,61].

### 2.6. Quantitative Real-Time PCR

Total RNA was extracted from cells using the RNeasy mini kit (QIAGEN #74004). Reverse transcription was carried out using the qScript cDNA Synthesis Kit (#95047 Quanta bio, Beverly, USA). The quantitative real-time PCR (qRT-PCR) experiments were performed using SYBR-Green reagents (Takara Bio Inc. Kusatsu, Shiga, Japan) on an Agilent Mx3005P system (Agilent Technologies, Inc., Santa Clara, CA, USA). Each sample was run in triplicate. Human BAG3 forward primer, 5′-AGCTCCGACCAGGCTACATT-3′ and reverse primer, 5′-GGATAGACATGGAAAGGGTGC -3′; LITAF forward primer, 5′-TCCTTCGTATTATACCCAGCCA-3′ and reverse primer, 5′-GTGCTGCACGTAGACCGTC-3′; CCl2 forward primer 5′-ATGAAAGTCTCTGCCGCCCTTCTGT-3′ and reverse primer 5′-AGTCTTCGGAGTTTGGGTTTGCTTG-3′. The expression was quantified using the 2^−ΔΔCt^ method.

### 2.7. Macrophage Migration Assay: A. “Wound Healing Assay”

THP1 was plated in a 12-well plate (1 × 10^6^ cells/well), differentiated with 100 ng/mL PMA and transfected with corresponding siRNAs for 72 h, treated with 5 µM JG98, 100 ng/mL CSF1, or 100 nM ROCK inhibitor Y-27632 for 24 h or left untreated, then the cell monolayer was scratched using a p10 tip, and “wound healing” was recorded 24 h later. For quantification, pictures of three random fields along the scratch were taken, and identical rectangles with a width corresponding to the width of the original scratch were drawn in these fields. Cells migrated into the areas of these rectangles were counted, and data were normalized to the number of cells migrated in control scratch.

### 2.8. Transwell Assay

Macrophages were plated (3 × 10^4^ cells/well) on a transwell insert (8 μM pore size) of a 24-well plate in 200 μL RPMI supplemented with 1% FBS with or without LITAF silencing. The bottom chamber was filled with the same media containing H1975 NSCLC cells supplemented with 10% FBS. Migrated cells were counted under a microscope, and data were normalized to the number of cells migrated in the control transwell. 

### 2.9. In Vitro Cell Labeling

A total of 2 × 10^7^ THP1 WT or LITAF or BAG3 silenced cells were labeled using the cytoplasmic membrane dyes HCS CellMask™ Deep Red Stain 650/655 nm (Thermo fisher #H32721) or CellBrite@NIR750 748/780 nm (Biotium Fremont, USA #30077), respectively, following the manufacturer’s staining protocol. Cells were suspended in PBSx1 or complete medium respectively at a density of 1 × 10^6^ cells/mL in 10 mL. Then, 1.5 μL of CellMask or 1 µl of CellBrite was added to each mL of the cells in suspension and thoroughly mixed by flicking the tube. After 90 min of incubation at 37 °C, the labeled suspension was centrifuged for 5 min at 1200 rpm. The supernatant was discarded, and cells were gently resuspended in 1 mL of pre-warmed PBSx1. The washing procedure involving centrifugation and resuspension in PBSx1 was repeated two more times before using the cells.

### 2.10. Mice and Tumors

For tumor xenografts, H1975 NSCLC cells were mixed at a 1:1 ratio with Matrigel, and 1 × 10^6^ cells were injected s.c. into both left and right flanks of 9 female NCR nude mice (Taconic). A total of 1 × 10^7^ THP1 WT and LITAF or BAG3 silenced cells or PBSx1 were injected intravenously twice on days 8 and 9. Image analysis was done after 24 h, i.e., day 10. Tumor growth was monitored using a caliper and calculated according to the formula *L* × *W*^2^ × π/6, where *L* is length and *W* is width. Excised tumors from mice were frozen at −80c with an O.C.T freezing compound and sliced using a cryostat.

### 2.11. Statistical Analysis

A statistical study using the R programming language was done. Data are presented as mean ± SD. The statistical significance of the difference between two or more groups was assessed using Student’s *t*-test and two-way analysis of variance (ANOVA), respectively. A *p*-value of 0.05 or less was regarded as significant. The quantitative analysis of each experiment was repeated 3 times (*n* = 3) to have statistical significance.

## 3. Results

### 3.1. Hsp70–Bag3 Regulates Levels and Activity of LITAF

As noted in the Introduction, we hypothesized that LITAF may link Hsp70 and Bag3 to the macrophage function, including tumor infiltration. Previously, LITAF was identified as one of the Bag3-interacting proteins in multiple high-throughput proteomics experiments (https://thebiogrid.org/114893/summary/homo-sapiens/litaf.html, accessed on 12 August 2022). Furthermore, a yet unpublished Ph.D. thesis demonstrated that such interaction occurs via the WW domain of Bag3 [59]. Therefore, we decided to reproduce these data in a direct co-IP experiment. His-tagged full-length Bag3 (FL) and ΔWW (deletion of WW domain) versions were expressed in 293T cells via lipofectamine transfection. Bag3 polypeptides were pulled down using Co^2+^-beads, and associated proteins resolved on SDS PAGE and probed with anti-LITAF antibodies in immunoblotting. As seen in Appendix A, lower amounts of LITAF were pulled down with ΔWW compared to the FL Bag3, strongly suggesting that, in fact, LITAF interacts with Bag3, and this interaction is mediated by the WW domain.

To test whether Bag3–LITAF interaction is functionally significant, we tested whether silencing of Bag3 affects levels of LITAF in macrophages. As a model, we used differentiated macrophage-like THP1 cells [62]. Accordingly, Bag3 was silenced using two different siRNAs, and levels of LITAF were assessed by immunoblotting after 48 h of silencing. Figure 1A shows that Bag3 silencing led to a significant downregulation of LITAF levels. Similar effects were seen upon incubation of cells with the Hsp70 inhibitor JG-98 [29], indicating that Hsp70 is also involved in LITAF regulation (Figure 1B). Therefore, it appears that LITAF levels are controlled by both components of the Hsp70–Bag3 complex. Considering the role of LITAF in the transcription of certain cytokines in macrophages, these data were in line with the idea that Bag3 and LITAF may link Hsp70 to the macrophage function.

To further explore this possibility, we tested the effects of Bag3 on the expression of the LITAF target CCL2, which plays a major role in tumor development [50]. As shown previously, silencing of LITAF led to a significant downregulation of CCL2 mRNA levels (Figure 1C), and similar effects were seen upon silencing of Bag3 (Figure 1C) or incubation of cells with JG-98 (Figure 1D). 

Recently, we uncovered that Bag3 directly interacts with another transcription factor, a major regulator of cell polarity and cancer development—Yap [59]. Via this interaction, Bag3 controls phosphorylation and nuclear localization of Yap. Furthermore, upon stimulation, Bag3 co-migrated with Yap to the nucleus [59]. Accordingly, we suggested that Bag3 may play a more general role in transcription by the regulation of nuclear localization of certain transcription factors, including LITAF. Therefore, we assessed how the silencing of Bag3 can affect the nuclear localization of LITAF by immunofluorescence of cells with LITAF antibodies. Figure 2 shows that there was very high variability in the ratio of nuclear/cytoplasmic LITAF in the population of differentiated THP1 cells. Though the silencing of Bag3 led to the downregulation of LITAF, the ratio of nuclear/cytoplasmic LITAF did not change. Therefore, the effects of Bag3 are limited to the levels of LITAF and are not related to the nuclear transport of this transcription factor.

### 3.2. BAG3 Affects Levels of LITAF via Chaperone-Mediated Autophagy, while Hsp70 Controls LITAF via Its Proteasome Degradation

Our next question was how Bag3 and Hsp70 control levels of LITAF. We tested possible ways of downregulation of LITAF, including downregulation of mRNA expression, acceleration of proteasome-dependent degradation of LITAF polypeptides, and acceleration of its autophagic degradation. Accordingly, we silenced Bag3 with the corresponding siRNA and measured mRNA levels of both Bag3 and LITAF. Figure 3A shows that while Bag3 was successfully silenced, LITAF mRNA levels were not significantly changed. Therefore, Hsp70–Bag3 could potentially affect events downstream of mRNA levels, i.e., either translation of LITAF or its degradation. Considering that Bag3 directly interacts with LITAF (Appendix A), the latter possibility seemed more likely. Accordingly, we assessed the degradation of LITAF upon silencing Bag3. We silenced Bag3, inhibited the proteasome activity with MG132, and measured LITAF levels by Western blot. In naïve cells, LITAF was degraded via the ubiquitin-proteasome system since the addition of the proteasome inhibitor MG132 significantly increased the levels of LITAF (Figure 3B). However, upon silencing of Bag3 addition of MG132 did not restore the reduced levels of LITAF, indicating that the drop of LITAF was not associated with the enhancement of its proteasome-dependent degradation.

Interestingly, in contrast to inhibition of the proteasome, inhibition of the lysosomal degradation by chloroquine or NH_4_Cl partially restored the levels of LITAF (Figure 3C), suggesting that silencing of Bag3 leads to acceleration of the autophagic degradation of LITAF. On the other hand, the addition of these inhibitors also partially restored the levels of Bag3 (Figure 3C), which complicated the interpretation of the results. Further experiments, however, clarified the role of autophagy in Bag3 effects on LITAF levels.

Since Bag3 mediates macroautophagy [63,64], we originally suggested that this pathway is involved in LITAF degradation. However, if Bag3 serves a function in the macroautophagic degradation of LITAF, one expects to see stabilization and accumulation of the latter upon silencing of Bag3 (as was seen with other autophagic substrates) [65,66]. In contrast, we saw destabilization of LITAF upon silencing of Bag3 (Figure 3C), suggesting that, although degradation of LITAF upon Bag3 silencing involves lysosomes (is inhibited by chloroquine and NH_4_Cl), it cannot be macroautophagy-dependent. Therefore, we suggested that under these conditions, LITAF is degraded via a chaperone-mediated LAMP2-dependent autophagy (CMA), upon which cytoplasmic protein molecules are directly injected into the lysosome compartment. To test this possibility, we silenced LAMP2 by siRNA (Appendix A) and assessed levels of LITAF upon silencing Bag3 (Figure 3D). Indeed, blocking CMA completely restored the levels of LITAF in Bag3-silenced cells. Therefore, silencing of Bag3 facilitates degradation of LITAF via the chaperone-mediated autophagy. This finding seemingly contradicted the fact that inhibition of Hsp70 by JG-98 caused a similar drop in the LITAF levels (Figure 1B) since Hsp70 facilitates the LAMP2-dependent autophagy. Therefore, we suggested that the pathway of LITAF degradation upon incubation of cells with JG-98 could be different. Indeed, the level of LITAF in the presence of JG98 was not restored by inhibition of lysosomal degradation (Figure 3E). In contrast, the drop in LITAF levels upon the addition of JG-98 was reversed almost completely by the addition of MG132 (Figure 3F), indicating the role of the proteasome-dependent degradation. Therefore, there are two pathways of LITAF degradation, one of which is stimulated by the silencing of Bag3, and another one by inhibition of Hsp70. 

### 3.3. Bag3 and LITAF Mediate Macrophage Migration and Tumor Infiltration by Controlling CSF-1 Pathway

In previous work, we showed that inhibition of Hsp70 leads to inhibition of macrophage migration and tumor infiltration [30]. Here, we addressed if other components of the Hsp70–Bag3–LITAF axis exert similar effects on macrophages. Indeed, scratch assay clearly demonstrated that silencing of either LITAF or Bag3 using the corresponding siRNA significantly suppressed macrophage motility (Figure 4A). Similarly, LITAF dramatically reduced macrophage infiltration in the invasion assay (Figure 4B), indicating that LITAF and Bag3 play a critical role in macrophage movement and invasiveness. 

To further explore this hypothesis, we tested the effect of Bag3 and LITAF on the macrophage’s ability to infiltrate tumors in vivo. We established subcutaneous xenograft tumors of MDA-MB231 cells in nude mice. Bag3 or LITAF were silenced in differentiated THP1 cells. Control cells were labeled with Cy5 (red) life color, while cells with silenced proteins with Cy7 (magenta) life color (Figure 4C,D). In vitro labeling followed by incubation without the label demonstrated that cells lose both dyes at similar rates so that in 24 h fluorescence diminishes by about 30% (Appendix A). Control and LITAF-silenced cells were mixed in equal proportions (Figure 4C,D) and injected into the tail vein of tumor-bearing mice. After 24 h, animals were sacrificed and imaged. Tumors were resected, sliced, and imaged by fluorescence microscopy. 

Figure 4E,F show that control cells (red) are distributed in the body of animals and accumulated in the tumor. In contrast, LITAF-silenced cells (magenta) were in the body but to a much lesser extent in the tumor. Tumor sections confirm that a much lower number of the LITAF-silenced cells infiltrated into the tumor, compared to control cells (Figure 4E,G). Similar results were obtained with Bag3-silenced cells (Figure 4F,H). Altogether, these data indicate that silencing of either Bag3 or LITAF significantly reduces the motility of macrophages and their infiltration into tumors.

### 3.4. Hsp70–Bag3–LITAF Axis Regulates Macrophage Motility via Controlling the Expression of CSF1

We further addressed how the Hsp70–Bag3–LITAF axis regulates motility and tumor infiltration of macrophages. THP1 macrophages were silenced of Bag3 or LITAF using the corresponding siRNA or treated with JG-98 for 24 h, mRNA was isolated from the control and treated cells, and global gene expression analysis was assessed by RNAseq. As expected, silencing of Bag3 and incubation with JG-98 caused similar changes in gene expression while changes caused by silencing of LITAF were more distinct. Nevertheless, there was a significant overlap (Appendix A). Analyzing these changes, we found a series of genes that are involved in macrophage motility to be downregulated by all three treatments (Figure 5A), including MMP9 [67], FOXM1 [68], MARCO [69] and CSF1 [40] (Additional common downregulated and upregulated genes can be found in Appendix A). CSF1 (MCSF) was of particular interest because of its pleiotropic effects on macrophage growth, survival, proliferation, differentiation, and motility [40,70]. We decided to test whether the supplement of CSF1 can restore the motility defects caused by the silencing of Bag3 and LITAF. In this regard, we silenced Bag3 or LITAF in differentiated THP1 cells and performed a wound healing assay with or without supplement of recombinant CSF1. In naïve cells, the addition of CSF1 did not significantly affect wound healing (Figure 5B,C). However, CSF1 significantly restored motility suppressed by the silencing of either Bag3 or LITAF (Figure 5B,C). 

To further explore this possibility, we tested if blocking the Hsp70–Bag3–LITAF axis affects the pathway downstream of CSF1. Since the CSF1 receptor controls the Akt pathway [40,70], we measured activation of Akt in cells treated by JG-98 or silenced Bag3 or LITAF. Indeed, all three treatments led to a strong suppression of the Akt phosphorylation at S473, indicating suppression of the kinase activity (Figure 5D). 

In regulating motility, Akt phosphorylates ROCK, which in turn phosphorylates MLC and LIMK [71], exerting negative regulation of cell movement [71,72,73,74]. ROCK was shown to be inhibited upon CSF1 activation [70]. Small molecules that inhibit ROCK have been developed [75]. Accordingly, we predicted that inhibition of ROCK with small molecule should restore macrophage motility suppressed by inhibition of the Hsp70–Bag3–LITAF axis. 

Indeed, inhibition of ROCK with Y-27632 significantly restored motility of differentiated THP1 cells suppressed by JG-98 or siBag3 or siLITAF (Figure 5D). These findings indicate that the CSF1 pathway is intricately involved in the regulation of macrophage motility by the Hsp70–Bag3–LITAF axis.

## 4. Discussion

In cancer treatment, there have been attempts to target TAMs, which reduces the chances to acquire drug resistance due to the selection of resistant clones [20]. Previously, we reported that targeting Hsp70 in the stroma can suppress tumor development due to the inhibition or infiltration of macrophages into the tumor site. These effects were seen in certain mice cancer models, such as allografts of E0771 breast cancer or B16 melanoma, while with other models, e.g., PyMT-induced breast cancer [21], the requirement for TAM was less strict. Overall, this work suggested that inhibitors of Hsp70 may effectively target tumor stroma.

Here, we addressed the mechanism of effects of Hsp70 on macrophage motility and their infiltration into tumors. Our starting point was the observation that JG-98 that suppressed tumor infiltration of macrophages can block the interaction of Hsp70 with Bag3 [26], suggesting that Bag3 can be involved in this effect. Furthermore, published data indicated that Bag3 can interact with a transcription factor LITAF that regulates many macrophage functions [76]. These considerations allowed us to understand the major stages of the pathway that links Hsp70 and macrophage motility. 

Overall, we found that Hsp70–Bag3 regulates the levels of LITAF by controlling its degradation, while LITAF controls macrophage motility via the regulation of expression of the chemokine CSF1. Interestingly, silencing of Bag3 and inhibition of Hsp70 stimulated LITAF degradation via different pathways, chaperone-mediated autophagy, and proteasome, respectively. Inhibition of Hsp70 by JG-98 dissociates Hsp70–Bag3, and Bag3 probably remains complex with LITAF. We suggested that this should facilitate autophagic degradation of LITAF since Bag3 is known to target proteins to the autophagic vacuole [77]. However, under these conditions, LITAF degradation was inhibited by MG132, suggesting that Bag3 can target LITAF to the proteasome. In line with this suggestion, silencing of Bag3 blocked the proteasome-dependent degradation of LITAF, while stimulating its Lamp2-dependent chaperone-mediated autophagy. Since the latter is dependent on Hsp70, we suggest that silencing of Bag3 allows the association of LITAF with Hsp70, which facilitates its interaction with Lamp2. These interactions seem to be very complex, and additional study is necessary to dissect the exact mechanisms of the process.

Bag3 has been implicated in the regulation of many signaling pathways [10]. Recently, we reported that it can directly interact with a transcription factor Yap, a component of the Hippo pathway, and control its nuclear localization [59]. There are similarities between associations of Bag3 with Yap and LITAF. For example, Bag3 directly interacts with kinase LATS1/2, which phosphorylates Yap [63]. Bag3 also directly interacts with kinase p38, which phosphorylates LITAF [57]. Accordingly, we originally suggested that Bag3 may control the nuclear localization of the transcription factor LITAF. However, direct measurements indicated that this is not the case, and the entire regulation is related to the degradation of LITAF.

Novel anticancer TAM-targeting therapies are constantly being developed and improved, which are expected to break through traditional tumor-associated therapies and gain favorable clinical treatment results. Furthermore, a combination of TAM-targeting drugs with other anticancer drugs to get better efficacy will be an irresistible trend [78].

We found that LITAF is involved in the regulation of macrophage motility and tumor invasion. Indeed, silencing of LITAF significantly suppressed macrophage motility and invasion in vitro, as well as their tumor infiltration in vivo. It appears that the major factor that links LITAF with macrophage motility is CSF1. Indeed, the supplement of exogenous CSF1 almost completely reversed the motility defect caused by silencing of either LITAF or Bag3. Downstream components of the CSF1 pathway were also downregulated upon silencing of either Bag3 or LITAF, and restoration of the downstream pathway by ROCK inhibitor partially suppressed the motility defect. Therefore, altogether we dissected a novel pathway of regulation of macrophage motility and infiltration to tumors.

## 5. Conclusions

In this study, we found that Bag3 and LITAF are key contributors to tumor macrophage infiltration which is mediated by the CSF1 pathway. In this regulation, Bag3 directly binds to LITAF and modulates its proteasome-dependent and chaperone-mediated autophagic degradation. Overall, targeting the Hsp70–Bag3–LITAF axis may be beneficial in suppressing tumor progression.

## Figures and Tables

**Figure 1 cancers-14-04168-f001:**
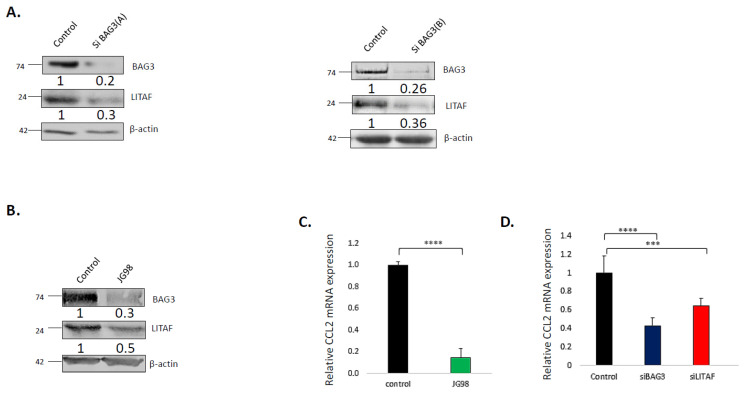
Effect of BAG3 on LITAF expression and activity. (**A**) Silencing of Bag3 downregulates LITAF. Differentiated THP1 macrophages were transfected with BAG3 or control siRNAs, as indicated. After 72 h, levels of indicated proteins were detected by immunoblot analysis. (**B**) Inhibition of Hsp70 leads to downregulation of LITAF. Cells were treated with 5 µM JG98 or left untreated. After 24 h, levels of indicated proteins were detected by immunoblot analysis. Numbers under the blot lanes represent relative expression normalized to β-actin. (**C**) Silencing of Bag3 causes downregulation of the LITAF target gene CCL2. THP1 cells were transfected with indicated siRNAs for 72 h followed by real-time PCR analysis of CCl2 mRNA levels. (**D**) Inhibition of Hsp70 causes downregulation of CCL2. THP1 cells were treated with 5 µM JG98 for 24 h followed by real-time PCR analysis of CCl2 mRNA levels using a two-sample *t*-test. Depicted is the mean relative expression (log2) ± s.e.m. *** *p* ≤ 0.001 and **** *p* ≤ 0.0001. Uncropped WB Appendix A#.

**Figure 2 cancers-14-04168-f002:**
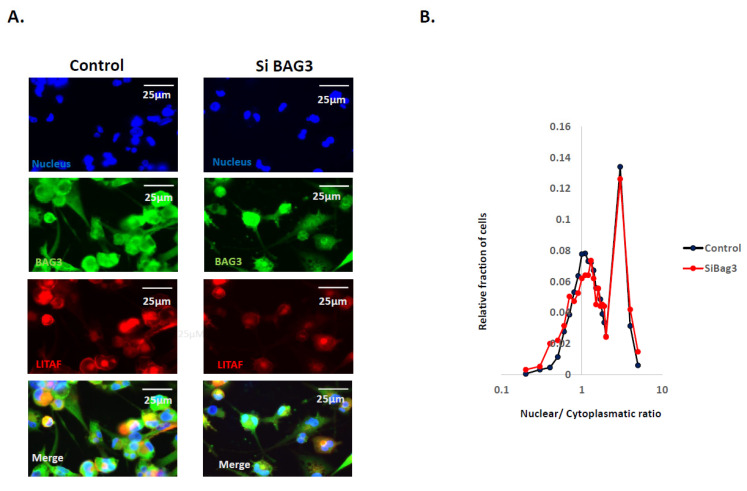
Effect of BAG3 on nuclear localization of LITAF. (**A**) Representative image of localization of LITAF in control and BAG3-silenced differentiated THP1 cells (images are representative of three experiments). (**B**) Distribution of cells in the population according to the N/C ratio (nuclear/cytoplasmatic) of LITAF in control cells and BAG3 silenced cells. Cells were treated on 96-well plates and fixed, and the localization of LITAF was assessed by immunofluorescence. Image acquisition and analysis were performed using the Hermes imaging system. Although LITAF levels were affected by the silencing of BAG3, the nuclear/cytoplasmatic ratio remained the same in control and BAG3 silenced cells.

**Figure 3 cancers-14-04168-f003:**
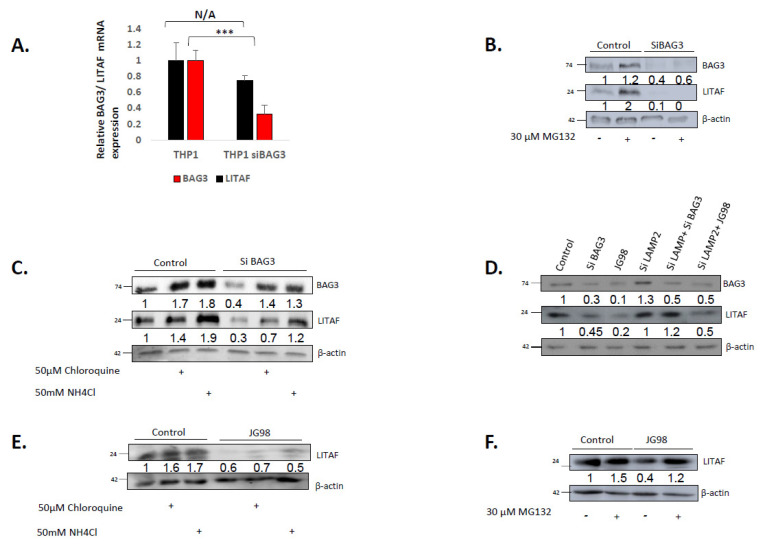
Mechanism of downregulation of LITAF by disruption of Hsp70–Bag3. (**A**) Silencing of Bag3 does not affect mRNA levels of LITAF. Differentiated THP1 macrophages were transfected with BAG3 or control siRNAs for 72 h followed by real-time PCR analysis of LITAF mRNA levels. mRNA levels were calculated using two-sample *t*-test statistics. Depicted is the mean relative expression (log2) ± s.e.m. *** *p* ≤ 0.001 and N/A *p* ≥ 0.05. (**B**) Inhibition of proteasome does not restore levels of LITAF upon silencing of Bag3. THP1 was transfected with the corresponding siRNAs for 72 h, followed by incubation with 30 µM MG132 for an additional 24 h, levels of indicated proteins were detected by immunoblot analysis. (**C**) Inhibition of lysosome significantly restores the levels of LITAF upon silencing of Bag3. THP1 was transfected with the corresponding siRNAs for 72 h, followed by incubation with 50 µM chloroquine for an additional 4 h or 50 mM NH4Cl for 30 min. Levels of indicated proteins were detected by immunoblotting. (**D**) Upon Bag3 silencing LITAF is downregulated via the CMA. THP1 were transfected with Bag3, Lamp2a, or control siRNA or their combinations for 72 h. Alternatively, cells were treated with 5 µM JG98 for 24 h in the presence and absence of Lamp2 siRNA, and the levels of indicated proteins were detected by immunoblot analysis. While silencing of Lamp2 restores levels of LITAF upon Bag3 silencing, it does not affect LITAF levels upon inhibition of Hsp70. (**E**) Inhibition of lysosome did not restore levels of LITAF upon downregulation of Hsp70 by JG98 (**F**) Inhibition of proteasome restores LITAF levels upon inhibition of Hsp70. Cells were treated with 5 µM JG98 or left untreated for 24 h, followed by incubation with 30 µM MG132 for an additional 24 h, levels of indicated proteins were detected by immunoblot analysis. Numbers under the blot lanes represent relative expression normalized to β-actin. Uncropped WB Appendix A#.

**Figure 4 cancers-14-04168-f004:**
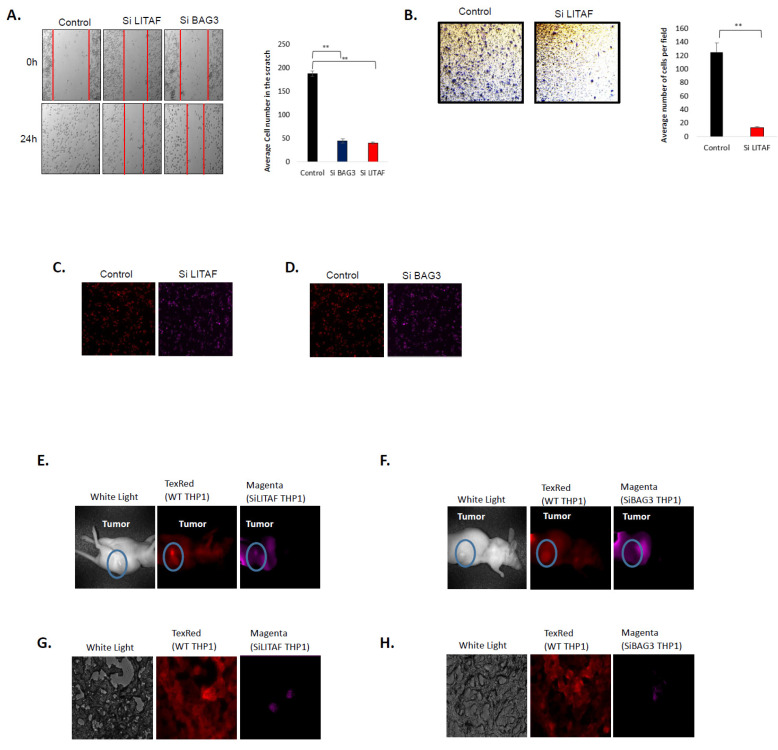
Silencing of Bag3 or LITAF suppresses motility and tumor infiltration of macrophages. (**A**) Silencing of Bag3 or LITAF suppresses cell motility in the wound healing assay. Differentiated THP1 macrophages were transfected with indicated siRNAs for 72 h and their migration was monitored by the “wound healing” assay. On the left—a representative image of cells migrated in wound healing assay, on the right (bar graph)—represents the quantification of the data. Cell numbers among the conditions were measured via Student’s *t*-test. (**B**) Silencing of Bag3 or LITAF suppresses cell invasion in the transwell assay. On the left—a representative image of cells migrated in transwell assay; on the right (bar graph)—demonstrates the quantification of the data using Anova. Data shown are means+/SEM of triplicates. ** *p* ≤ 0.01. (**E**–**H**) Silencing of LITAF or Bag3 suppresses the infiltration of macrophages in a mouse tumor. Cells were transfected with either Bag3 or LITAF or control siRNAs and labeled with red life color (control cells) or magenta life color (Bag3-silenced or LITAF-silenced cells). An equal number of control and silenced cells were mixed (**C**,**D**). Nine nude mice were first injected s.c with 106 H1975 NSCLC cells in Matrigel to form xenograft tumors. On days 8 or 9 post-injection, when tumors were formed, mice were intravenously injected with the mixture of control and LITAF-silenced (**E**) or BAG3 silenced (**F**), or PBSx1 (picture not shown) red or magenta stained differentiated THP1 macrophages. On day 10 tumors were collected and sliced and pictures were taken using a fluorescent microscope (**G**,**H**). Whole mouse images of the experiment are described in C and D. Image of control (red) or siLITAF (magenta) (**E**) or siBAG3 (magenta) (**F**) THP1 distribution in the body of the mouse.

**Figure 5 cancers-14-04168-f005:**
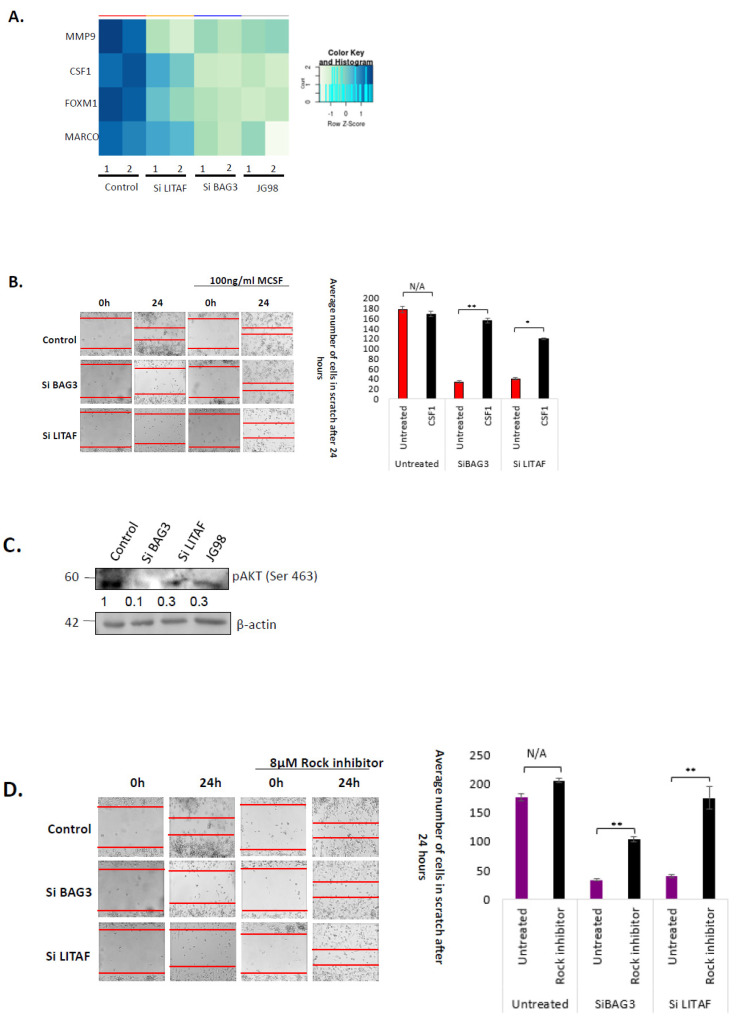
Effects of Bag3 and LITAF on macrophage motility are mediated by CSF1. (**A**) Effects of silencing of Bag3 or LITAF or inhibition of Hsp70 on the expression of motility-related genes. RNA was isolated from macrophages that were treated with siBAG3, siLITAF, control siRNA, or 5 µM JG98, and gene expression was assessed by RNAseq. Genes related to motility/migration are shown. The figure presents data from two independent experiments. Color coding shows a False Discovery Rate (FDR). (**B**) The addition of CSF1 significantly restores cell motility upon silencing of Bag3 or LITAF. Differentiated THP1 macrophages were transfected with indicated siRNAs for 72 h, followed by incubation with 100 ng/mL CSF1. The bar graph on the right represents the calculation of the number of cells. Quantification was done via Student’s *t*-test statistic. ** *p* < 0.01, * *p* < 0.5. (**C**) Silencing of Bag3 or LITAF or inhibition of Hsp70 caused dephosphorylation of CSF1 downstream target Akt Ser463. Differentiated THP1 macrophages were transfected with BAG3, LITAF, or control siRNAs for 72 h or treated with 5 µM JG98 for 24 h. Levels of indicated proteins were detected by immunoblot analysis. (**D**) Inhibition of the Akt target ROCK restores cell motility upon silencing of Bag3 or LITAF. A total of 8 µM ROCK inhibitor was incubated with cells for 24 h and macrophage migration was monitored in the “wound healing” assay. (**Left panel**) assay as described in Figure 5B. **Right panel**—quantification of the migration data using Anova. Data shown are means+/SEM of triplicates. ** *p* ≤ 0.01. Uncropped WB Appendix A#.

## Data Availability

Materials described in the manuscript, including all relevant raw data, will be freely available to any researcher wishing to use them for non-commercial purposes, without breaching participant confidentiality. RNAseq data can be found here.

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
