# Peer review of "Hsp70–Bag3 Module Regulates Macrophage Motility and Tumor Infiltration via Transcription Factor LITAF and CSF1"

_cancers, 2022, doi:10.3390/cancers14174168_

Round 1

Reviewer 1 Report

The article by Avinery and colleagues discusses the role of HSP70-Bag3-LITAF-CSF1 pathway in controlling the macrophage motility and tumor infiltration. The article is well crafted, executed and the conclusions drawn are supported by the results. However, I do have several comments before publication is possible. 

1.     Be consistent in usage of acronyms, inhibitors, Si RNA’s and italicize the gene names following international nomenclature.

2.     In the introduction section expand on the role of CSF1 in tumor progression with necessary literature.

3.     In the methods section clearly indicate the number of cells used for each assay in respective methods.

4.     Section 2.9 mention number of mice used in the in vivo studies.

5.     In most of the figure authors have presented with significance but failed to mention anywhere in the manuscript how the statistical analysis was done. This should be presented in the revised version for better reproducibility.

6.     Axis missing in the figure 1 C and D, what does these represents? 

7.     Quantify all the Westerns presented in the manuscript with bar graphs.

8.     Figure 2A, provide the composite/merge image along with brightfield images. Also provide the scale bar in each of microscopy images. Axis missing in figure 2B.

9.     How authors claim downregulation of LITAF by disruption of Hsp70-BAG3 and not by interacting with other proteins? Do they include any positive controls?

10.  Authors should mention the inclusion of chloroquine in the methods section for detecting lysosomal/proteasomal degradation assays and not directly dwelling to the results section without proper methodology section.

11.  Figure 3A axis missing.

12.  Authors should include a conclusion paragraph.

Author Response

Dear Reviewer.

Thank you very much for handling our manuscript and for the comments. We addressed all points raised by you, and hope that in the present form our paper is suitable for publication. Below please find point-by-point response to the comments:

  1. Be consistent in usage of acronyms, inhibitors, Si RNA’s and italicize the gene names following international nomenclature.

We fixed the text accordingly.

  1. In the introduction section expand on the role of CSF1 in tumor progression with necessary literature.

We expanded Introduction, as requested.

  1. In the methods section clearly indicate the number of cells used for each assay in respective methods.

Done

  1. Section 2.9 mention number of mice used in the in vivostudies.

Done

  1. In most of the figure authors have presented with significance but failed to mention anywhere in the manuscript how the statistical analysis was done. This should be presented in the revised version for better reproducibility.

Info about statistical analysis is introduced in the caption to Fig. 1.

  1. Axis missing in the figure 1 C and D, what does these represents? 

Fixed

  1. Quantify all the Westerns presented in the manuscript with bar graphs.

We prefer to keep our original quantification as a number under a Western blot lane. Bar graph will require averaging over different repeats, which will artificially create large deviations.

  1. Figure 2A, provide the composite/merge image along with brightfield images. Also provide the scale bar in each of microscopy images. Axis missing in figure 2B.

Done

  1. How authors claim downregulation of LITAF by disruption of Hsp70-BAG3 and not by interacting with other proteins? Do they include any positive controls?

We only claim that disruption of Hsp70-Bag3 leads to downregulation of LITAF. This effect is probably direct since Bag3 directly interacts with LITAF. We are not arguing that other proteins are not involved in the process. In fact, we describe that LAMP2 is also involved in downregulation.

  1. Authors should mention the inclusion of chloroquine in the methods section for detecting lysosomal/proteasomal degradation assays and not directly dwelling to the results section without proper methodology section.

Done

  1. Figure 3A axis missing.

Done

  1. Authors should include a conclusion paragraph.

Done

Reviewer 2 Report

The study demonstrates that the HSP70-Bag3-LITAF-CSF1 pathway controls macrophage motility and tumor infiltration. Co-chaperone Bag3 links Hsp70 and regulate expression of inflammatory cytokines and chemokines in macrophages via transcription factors LITAF and CSF1.

The mechanism how NH4Cl restores the levels of LITAF may be discussed more in detail in terms of Figure 3C. Discussion may be revised to specify the cancer treatment that target TAMs. Please check the reference 21 carefully.

Author Response

Dear Reviewer.

Thank you very much for handling our manuscript and for the comments. We addressed all points raised by you, and hope that in the present form our paper is suitable for publication. Below please find point-by-point response to the comments:

The study demonstrates that the HSP70-Bag3-LITAF-CSF1 pathway controls macrophage motility and tumor infiltration. Co-chaperone Bag3 links Hsp70 and regulate expression of inflammatory cytokines and chemokines in macrophages via transcription factors LITAF and CSF1.

The mechanism how NH4Cl restores the levels of LITAF may be discussed more in detail in terms of Figure 3C. Discussion may be revised to specify the cancer treatment that target TAMs.

Done

Please check the reference 21 carefully.

Fixed

Round 2

Reviewer 1 Report

Authors have satisfactorily answered my initial comments.

Still it has few typos that has to be fixed before publication. Example figure 2A the scale bars should in meter (m) and not in molar (M). I would suggest the authors to include a statistical analysis part in the methods section for better readership.

Author Response

Dear Reviewer.

Thank you very much for handling our manuscript and for the comments. We addressed all points raised by you, and hope that in the present form our paper is suitable for publication. Below please find point-by-point response to the comments:

Still it has few typos that has to be fixed before publication. Example figure 2A the scale bars should in meter (m) and not in molar (M). I would suggest the authors to include a statistical analysis part in the methods section for better readership.

Both were fixed
